# Dual RNA Sequencing of *Mycobacterium tuberculosis*-Infected Human Splenic Macrophages Reveals a Strain-Dependent Host–Pathogen Response to Infection

**DOI:** 10.3390/ijms23031803

**Published:** 2022-02-04

**Authors:** Víctor A. López-Agudelo, Andres Baena, Vianey Barrera, Felipe Cabarcas, Juan F. Alzate, Dany J. V. Beste, Rigoberto Ríos-Estepa, Luis F. Barrera

**Affiliations:** 1Grupo de Inmunología Celular e Inmunogenética (GICIG), Instituto de Investigaciones Médicas, Facultad de Medicina, Universidad de Antioquia, Medellín 050010, Colombia; v.lopez-agudelo@ikmb.uni-kiel.de (V.A.L.-A.); andres.baenag@udea.edu.co (A.B.); 2Grupo de Bioprocesos, Facultad de Ingeniería, Universidad de Antioquia, Medellín 050010, Colombia; Rigoberto.rios@udea.edu.co; 3Programa de Ingeniería Biológica, Universidad Nacional de Colombia, Sede Medellín, Medellín 050010, Colombia; vpbarrerae@unal.edu; 4Grupo Sistemas Embebidos e Inteligencia Computacional (SISTEMIC), Facultad de Ingeniería, Universidad de Antioquia, Medellín 050010, Colombia; felipe.cabarcas@udea.edu.co; 5Centro Nacional de Secuenciación Genómica (CNSG), Sede de Investigación Universitaria (SIU), Facultad de Medicina, Universidad de Antioquia, Medellín 050010, Colombia; jfernando.alzate@udea.edu.co; 6Department of Microbial Sciences, Faculty of Health and Medical Science, University of Surrey, Guildford GU2 7XH, UK; d.beste@surrey.ac.uk

**Keywords:** human, splenic macrophages, *Mycobacterium tuberculosis*, clinical strains, dual RNA-seq, metabolic reconstruction

## Abstract

Tuberculosis (TB) is caused by *Mycobacterium tuberculosis* (Mtb), leading to pulmonary and extrapulmonary TB, whereby Mtb is disseminated to many other organs and tissues. Dissemination occurs early during the disease, and bacteria can be found first in the lymph nodes adjacent to the lungs and then later in the extrapulmonary organs, including the spleen. The early global gene expression response of human tissue macrophages and intracellular clinical isolates of Mtb has been poorly studied. Using dual RNA-seq, we have explored the mRNA profiles of two closely related clinical strains of the Latin American and Mediterranean (LAM) family of Mtb in infected human splenic macrophages (hSMs). This work shows that these pathogens mediate a distinct host response despite their genetic similarity. Using a genome-scale host–pathogen metabolic reconstruction to analyze the data further, we highlight that the infecting Mtb strain also determines the metabolic response of both the host and pathogen. Thus, macrophage ontogeny and the genetic-derived program of Mtb direct the host–pathogen interaction.

## 1. Introduction

The causative agent of tuberculosis (TB), *Mycobacterium tuberculosis* (Mtb), is a very unusual bacterial pathogen in that it can cause both an acute, severe active disease and persist for decades, causing a latent infection in the human host. The primary host cell harboring Mtb is the macrophage, the very cell designed to kill invading microbes, and therefore understanding the intricate cross-talk between Mtb and the macrophage has inspired numerous studies. Pulmonary TB is the most common form of active TB, explaining 85% of the total cases, while the remainder of TB is extrapulmonary due to the hematogenous spread of bacteria from the lungs to any site of the body [1,2]. However, it is increasingly recognized that all Mtb infections involve early dissemination out of the lungs into the neighboring lymph nodes and subsequently into the spleen [3]. However, few studies have investigated the cross-talk between Mtb and extrapulmonary macrophages.

In humans, most studies have focused on understanding the survival of the laboratory strain of Mtb H37Rv in monocyte-derived macrophages (MDMs) or monocytic cell lines. Seminal research has shown that macrophage ontology significantly impacts their environment for Mtb [4,5,6,7]. However, less work has been performed on understanding how the Mtb strain itself drives the host–pathogen interaction. Based on the current evidence in animal models of infection [8,9,10,11] and in vitro in mice [12,13,14,15,16] and human macrophages [17,18,19], the prevalent view states that genomic and transcriptomic differences among clinical isolates of Mtb can explain the distinct immune responses in the host. Furthermore, significant correlations between genotype and transcriptional phenotype of clinical isolates cultured in vitro have also been observed [20,21,22]. However, most referred studies have compared infection of macrophages with Mtb from different lineages, while comparison between closely related clinical strains has been poorly explored [16].

Genome-wide transcriptomics using RNA-seq provides a snapshot of the cell’s physiological state under the given condition. For example, mRNA profiling of mononuclear phagocytes infected in vitro with Mtb has highlighted the innate immune response’s complexity to mycobacterial infection [23,24,25,26,27]. Using microarray analysis, we previously explored the transcriptomic response of human alveolar and splenic macrophages to infection with clinical isolates of Mtb [28], which indicated Mtb strain-specific modulation of the infected macrophages and differences between the response of the two types of macrophages to infection. However, we did not explore the transcriptional response of Mtb to these intracellular environments in that study.

Dual RNA-seq provides a powerful approach to measuring the host and pathogen transcriptome [29]. When applied to the Mtb strain H37Rv growing within THP-1 macrophages, [30] discovered that Mtb accesses a wide range of host-derived substrates and profiled Mtb transcriptomic and metabolomics responses. This study was highly informative but with the caveat that the cancer cell line THP-1 has many de-regulated metabolic reactions. It is now well established that changes in infected cells’ metabolic response drive the immune response [31]. For example, accumulated evidence indicates that murine and human tissue M-1-like inflammatory macrophages infected with Mtb display enhanced glycolysis, pentose phosphate, and fatty acid synthesis, coupled with higher expression and activity of transcription factors such as HIF-1α. On the other hand, M2-like anti-inflammatory macrophages show reduced glycolysis, enhanced oxidative phosphorylation, and fatty acid oxidation [32].

Using the dual RNA-seq approach, we profiled the transcriptomic cross-talk between human splenic macrophages (hSMs) infected with two clinical isolates of the Latin American and Mediterranean (LAM) family of Mtb UT127 and UT-205. Our previous genomic characterization of those isolates demonstrated that the strains are highly genetically similar (99.6%) [22,33]; however, evidence that they induce a different pattern of cell death in human alveolar and monocyte-derived macrophages [34] and a distinct transcriptomic response in human alveolar and splenic macrophages [35] led us to the hypothesis that these clinical strains drive very different host responses to infection. Indeed, there was a more robust innate immune response to UT127 clinical isolate, suggesting an early and more potent recognition by the innate immune system than the UT205 strain, which may have consequences in the progression of the disease. Moreover, we found a mixture of M1/M2-like polarization in the splenic macrophages infected with the clinical isolates.

## 2. Results

The main limitation to collecting meaningful information on the gene expression pattern of Mtb inside the infected host cell is the reduced amount of RNA collected from infected cells. This limitation is even more substantial in humans, given the limited availability of tissue macrophages. To overcome these limitations, we used hSMs from spleen slices obtained from deceased transplant donors [28,34,36]. Under the present experimental conditions (Figure 1A), we infected 5–20 million macrophages with two closely related clinical isolates of the LAM family of Mtb, UT127, and UT205 [22,33]. To obtain enough RNA of the Mtb isolates for the dual RNA-seq experiments, we infected the macrophages with an MOI of 10:1, mimicking a condition of Mtb replication inside the macrophage. Total RNA was extracted and used for dual RNA sequencing six hours later. Our previous studies using hSMs under the same conditions described here showed that more than 85% of the cells were infected with either clinical strain [28,34]. Our interest was to examine an early transcriptional response of the infected macrophages and Mtb clinical isolates under the assumption that early transcriptional events may be critical for the host–pathogen interaction occurring later during infection. Our previous studies indicated that within 6 h of infection, no cell death of infected macrophages was observed despite the high MOI used [28,34], precluding the undesirable effects on the amount of RNAs recovered. Controls included non-infected hSMs and the clinical strains cultured in Sauton’s minimal medium.

There were no differences in the overall number of differentially expressed genes between hSMs infected with either of the two clinical strains (Appendix A). However, following our previous work, the principal component analysis showed that there was no clustering between the transcriptomic response of macrophages (PC2) infected with the two clinical strains, indicating a unique Mtb strain-driven response by the macrophages [28] (Appendix A). As expected, non-infected macrophages separated from Mtb-infected macrophages (PC1).

Similarly, PCA separated the transcriptomic responses of intracellular UT127 and UT205 Mtb and in vitro cultured Mtb strains into three clusters (Figure 1B). The intracellular transcriptomic responses of the two strains separated into two distinct clusters when growing ex vivo in splenic macrophages, indicating a unique response to the intracellular environment of hSMs. Whereas during in vitro growth, the transcriptomic responses were similar and clustered together (Figure 1B).

### 2.1. Infection with the Clinical Strains UT127 and UT205 Induced a Distinct Transcriptomic Response of Splenic Macrophages

Infection of hSMs with Mtb resulted in an extensive and complex gene expression response (Appendix A). Based on our selection criteria for differentially expressed genes (log_2_-fold = 1.5; *p*-value < 0.05; FDR < 0.05), hSMs up-regulated 972 genes in response to UT127 and 957 genes in response to UT205, from which 855 were commonly expressed while 117 were unique to infection with UT127 and 102 were unique to UT205. Infection with Mtb also resulted in a more extensive effect on downregulated genes in hSMs infected with UT127 (*n* = 1025), while 920 genes were downregulated upon infection with UT205; 841 were commonly downregulated, while 184 were unique to UT127 and 79 to UT205 (Appendix A).

To examine the complexity of the macrophage response to Mtb infection, we used the String database to find the significantly (FDR < 0.05) associated biological processes (BPs) with common and unique up-regulated and downregulated genes (Appendix A). Common up-regulated genes associated with 919 BPs while commonly downregulated genes associated with 93 BPs, indicating that the early infection with Mtb induced a substantial change in gene expression of hSMs. The top 20 common BPs of the up-regulated genes were mainly associated with the immune and inflammatory responses and response to cytokines, among others (Appendix A). On the other hand, the top 20 BPs associated with the downregulated genes were mainly associated with changes mostly with gene silencing and metabolic processes (Appendix A). Notorious differences were also observed between the BPs associated with the up-regulated genes of the hSMs infected with UT127 or UT205. While the response to the UT127 infection was enriched in BPs associated with metabolic and biosynthetic processes, lymphocyte and T cell differentiation, among others (Appendix A), the response of hSMs to UT205 infection was enriched in BPs associated with an immune response dominated by type I IFNs (Appendix A). Further, the BPs associated with the downregulated genes of hSMs to infection with UT127 were enriched in categories such as lipid and fatty acids metabolic processes, among others (Appendix A). In contrast, the infection with UT205 was associated with a more diverse enrichment and negative regulation that included categories of cell cycle regulation, DNA replication, and chromatin silencing, among others (Appendix A).

### 2.2. Splenic Macrophages Have a Unique Immune Response to Mtb Infection

To better understand the gene expression response of hSMs to infection with Mtb, we specifically focused our analysis on the immune response genes. For this analysis, the resulting complete list of DEGs was compared against the human immune genes of the ImmPORT database that collects immunological data from more than 10,000 healthy human individuals [37,38] (Appendix A) and then used GO-seq, a method for identifying differentially expressed gene sets, while accounting for the biases inherent to sequencing data such as gene length and read count [39] (Appendix A).

Common biological processes included the well-known innate response of macrophages to infection with Mtb, including cytokines and chemokines, activation of the JAK-STAT signaling and MAPK signaling pathways, response to hypoxia, myeloid cell differentiation, and granulocyte migration. However, GO categories included the regulation of IFN-γ production, chemotaxis, IFN-γ production, negative regulation of IL-10 production, regulation of JAK-STAT cascade, and myeloid cell differentiation, and regulation of the ERK1 and ERK2 cascade was differentially over-represented in response to UT127 as compared with UT205 infection. Conversely, cytokine production, innate immune response, response to hypoxia, and granulocyte migration GO categories were more prominent in response to UT205 than UT127 infection (Figure 1C). Interestingly, GO-seq analysis also allowed us to detect biological processes specifically enriched in hSMs in response to UT127 or UT205 infection (Figure 1D,E). Early response to infection with UT127 was mainly associated with T-cell responsiveness (Th1 and Th17) and positive myeloid cell differentiation (Figure 1D). In contrast, the macrophage response to UT205 infection was primarily characterized by negative regulation of cell death and the type I IFN response (Figure 1E).

As a complementary analysis, we used the signaling pathway impact analysis (SPIA) that combines classical enrichment analysis [40] with measurement of the actual perturbation in a given pathway under a given condition. This analysis showed an enrichment of Th1 and Th2 cell differentiation in response to UT127 infection and necroptosis cell death in hSMs infected with UT205 (Appendix A).

In summary, these results confirm and extend our previous observations showing that human tissue macrophages had a distinct response to the clinical strains [28]. More importantly, our data show that the infection with UT127 and UT205 strains results in a distinct transcriptomic effect associated with a differential immune response to the infection of hSMs.

### 2.3. M. tuberculosis UT127 and UT205 Display Distinct Transcriptomic Profiles Inside hSMs

Our dual RNA-seq analysis allowed us to map the Mtb expressed transcriptome to more than 80% of the Mtb genome (Figure 2A, see Section 4). Then, we analyzed the Mtb gene expression inside the hSMs taking the clinical isolates grown in Sauton’s minimal medium to control the clinical isolates grown in hSMs. This analysis identified 585 genes (14.9% of all genes) as differentially expressed by the Mtb UT127 strain; from these, 347 genes were up-regulated (59%), while 238 (41%) were downregulated. Similarly, 483 genes (12.3%) were identified as significant DEGs for the Mtb UT205 strain; from these, 291 were up-regulated (60%), while 192 (40%) were downregulated (Figure 2B,C) (Appendix A).

The most significant up-regulated genes in both strains were *prpD*, *prpC*, and *rv1057*. *prpD* (*rv1130*, methylcitrate dehydratase) and *prpC* (*rv1131*, methylcitrate synthase) are genes required for a functional propionyl CoA-assimilating methylcitrate cycle (MCC), and they are controlled by the transcriptional activator *prpR* (*rv1129c*), which also was up-regulated in both Mtb strains (Figure 2D,E). These genes are required for intracellular growth in macrophages and propionate metabolism in vitro [41,42], suggesting the utilization of propionyl-CoA yielding carbon sources such as sterols, odd-chain fatty acids, or branched-chain amino acids during infection of hSMs. Furthermore, *rv1057* is expressed at high levels in the early stages of Mtb infection of human MDMs [43]. Rv1057 participates in the secretion of the virulence factor ESAT-6 [44], inhibiting human T cells’ ability to produce IFN-γ, IL-10, and TNF-α [45]. Therefore, the upregulation of the *rv1057* gene in UT127 and UT205 seems to be a critical common immune strategy for surviving the initial adaptation stages inside hSMs.

To gain further insights, we performed an enrichment analysis of the common and unique up-regulated genes (Appendix A). From the commonly up-regulated genes, we identified 34 significant biological processes (FDR < 0.05), including response to hypoxia (FDR 0.00031) and cell wall biosynthesis (FDR 0.00031). Genes such as *rv0252* (*nirB*), *rv1736* (*narX*) and *rv1737c* (*narK2*) were highly upregulated (Figure 3A,B). A comparison of *dosR* regulon genes with known functions expressed by UT127 and UT205 shows that most of them were more highly induced in UT205 compared with UT127, including the transcriptional regulator devR (*rv3133c*) and the histidine sensor kinase DevS (*rv3132c*), as well as *narK2* (*rv1737c*), *narX* (*rv1736c*), *nrdZ* (*rv0570*), *pfkB* (*rv2029c*), *hspX* (*rv2031c*), *tgs1* (*rv3130c*) and *bfrB* (*rv3841*), suggesting a possible strategy response of the UT205 isolate to enter into a dormancy state in response to the stressful conditions posed by the macrophage (Appendix A). However, *dosR* genes such as *ctpF* (*rv1997c*), *fdxA* (*rv2007c*), and TB31.7 (*rv2623*) were more up-regulated in UT127 compared with UT205 (Appendix A). The *ctpF* gene encodes a probable metal cation transporter P-type ATPase A involved in calcium (Ca^2+^) efflux [46]. It is up-regulated in macrophages in response to hypoxia, high NO levels, and acidic pH, causing an alteration of the host cell calcium levels and diminished activation of mTOR, which was related to inhibition of autophagy and enhanced Mtb survival [47]. Ferredoxin A, and iron-sulfur-containing protein, is encoded by the gene *fdxA* and induced by hypoxia and low pH [48,49].

Then, we focused on the unique biological processes associated with the DEGs expressed by the clinical strains (Figure 4A–F). Biological processes such as the biosynthesis of siderophore group nonribosomal peptides pathway (FDR 9.13 × 10^−6^), transition metal ion homeostasis (FDR 0.0022), Catechol-containing compound metabolic process (FDR 0.0022), response to metal ion (FDR 0.0116), metal ion transport (FDR 0.0135), and response to copper (FDR 0.0155) and cadmium (FDR 0.0249) ions were identified as significantly altered in intracellular UT127 (Figure 4A–C). The response of Mtb to the metal cation assault of macrophages is critical for its survival, particularly to metal cations such as copper and cadmium, which are toxic in excess [50]. For example, animal models of infection with Mtb have shown that cupper transport proteins seem essential for copper resistance and Mtb survival [50,51]. Furthermore, iron acquisition from the host cell is vital for Mtb growth and survival. Mtb uses mycobactins and carboxymycobactins siderophores to obtain iron from the host cell. The fact that UT127 expresses higher levels of genes involved in the synthesis of mycobactins (*rv2383c/mbtB* and *rv2386c/mbtI*) and ferric iron-carboxymycobactins (*rv1348*/*irtA* and *rv1349*/*irtB*) may suggest a better capacity of this strain to grow in iron-limiting conditions compared with UT205.

In contrast, the cholesterol metabolic (FDR 1.62 × 10^−9^) and catabolic (FDR 2.65 × 10^−9^) processes and steroid metabolic process (FDR 6 × 10^−9^) were the top biological processes associated with the up-regulated genes expressed by UT205 (Figure 4D–F). Indeed, most of the genes belonging to the cholesterol metabolic process display a higher expression in UT205 compared with UT127 (Figure 4F). Cholesterol and fatty acids are used by Mtb and are associated with bacterial persistence and pathogenesis [52,53]. It has been proposed that the cholesterol-derived propionyl-CoA promotes an increase in the length and abundance of lipids such as PDIM and sulfolipid 1, both associated with Mtb virulence [53].

Intracellular UT127 and UT205 significantly downregulated a total of 331 genes. Of these, 99 (29.9%) were commonly downregulated, while 139 (42%) were unique to UT127, and 93 (28.1%) were unique to UT205 (Appendix A). A common cluster of genes belonging to the Mce/MlaD and MlaE superfamily proposed as lipid or steroid transporters and integrin-binding and required for persistence [54] are included in the downregulated genes. Our results show that 8 out of the 12 members of the *mce-1* (mammalian cell entry) operon (*rv0167*–*rv0174*) are included in this cluster of downregulated genes (Appendix A). It has been suggested that *mce* operon may be expressed at different times during infection [54]. Interestingly, functional inactivation of the *mce-1* operon confers hypervirulence in mice [55,56], impairs Th1 response [55], and inhibits TLR2 mediated pro-inflammatory responses in macrophages and epithelial cells [57]. More recently, it has been shown that disruption of the complete *mce-1* operon (13 genes), including *mmpL8*, *mmpL10*, *stf0*, *pks2,* and *papA2* involved in the transport and metabolism of lipids, led to a reduced expression of pro-inflammatory genes in macrophages [58]. UT127 expresses higher levels of mRNA for *mmpL8* (*rv3823c*), *mmpL10* (*rv1183*), *pks2* (*rv3825c*), and *papA2* (*rv3820c*), compared with UT205 (Appendix A), suggesting that infection with UT127 may induce a higher pro-inflammatory response compared with UT205. Thus, our observation is consistent with previous evidence showing that the *mce-1*operon genes mRNA and protein expression are repressed in infected macrophages early upon infection [59,60,61]. Besides, Oxidative phosphorylation (FDR 1.85 × 10^−9^) was the most significant gene network downregulated uniquely in intracellular UT127 (Appendix A). The *nuo* gene cluster encodes NADH dehydrogenases capable of oxidizing the cofactor NADH into NAD+ and plays an essential role in the mycobacterial respiratory chain [62]. The *nuo* gene cluster, particularly the *nuoG* gene (*rv3251c*), has been shown to induce anti-apoptotic activities and persistence in infected macrophages by neutralizing NOX-2 derived ROS and inhibiting apoptosis mediated by TNF-α [63,64,65]. Thus, it may be that during the early time we studied the mRNA expression (6 h), the UT127 did not interfere with the pro-apoptotic activities of the infected splenic macrophages. No significant biological processes or gene set networks were significantly associated with UT205 downregulated genes. However, contrary to UT127, a cluster of genes associated with the siderophore synthesis and iron acquisition is downregulated (Appendix A).

Taking together, our results indicate that both strains of Mtb respond to the intracellular environment of splenic macrophages by displaying distinct genetic strategies that may allow them to survive to the restrictive conditions imposed by the macrophage.

### 2.4. The Infecting Strain of M. tuberculosis Drives Distinct Metabolic Environments

Genome-scale metabolic models (GEMs) have become a valuable tool for analyzing omics datasets that remain challenging to interpret at a systems level without modeling. Therefore, we used a host–pathogen metabolic modeling approach to analyze our dual RNA-seq data further. We merged our Mtb genome-scale model (sMtb2.0) [66] with the human alveolar macrophage model [30] and the Integrative Metabolic Analysis Tool (iMAT) for the integration of transcriptomics data into genome-scale models [67]. This method assumes that RNA levels are correlated with metabolic fluxes. This method’s advantages are that the network imposes metabolic stoichiometry as an additional constraint to the transcriptomic data and that an objective function is not required [68]. Defining the objective function for pathogens such as Mtb is always problematic as it is unlikely to maximize the growth rate in vivo.

We mapped the differentially expressed hSMs and Mtb strains genes into the Host–Pathogen genome-scale model (Appendix A). Our simulations using iMAT predicted increased lactate production in UT205-infected hSMs and a corresponding increase in lactate consumption by Mtb UT205 (Appendix A). Lactate is a potential intracellular carbon source for Mtb [69]. Lactate is produced in larger quantities in human macrophages during aerobic glycolysis (the Warburg effect), which is also the hallmark of the M1-like phenotype, where cells increase energy demands (ATP production) for the production of inflammatory cytokines, nitric oxide (NO), reactive oxygen species (ROS), and prostaglandin [70,71]. Thus, Mtb UT205 drives an M1-like macrophage phenotype in hSM. By contrast, UT127-infected hSMs had a metabolic profile with features common to both M1-like and M2-like phenotypes (Figure 5A).

Turning to the intracellular pathogen, there were increased fluxes through glycolysis/gluconeogenesis, the citrate cycle, and mycolic acid pathway in Mtb UT127 compared with UT205. In contrast, cumulative fluxes through alanine, aspartate, and glutamic acid metabolism were higher for intracellular Mtb UT205 (Figure 5B). Similarly, iMAT predicted uptake consumption of 13 amino acids (D-alanine, L-proline, L-serine, L-leucine, L-isoleucine, L-asparagine, L-methionine, L-valine, glycine, L-tyrosine, L-phenylalanine, and L-threonine) by both Mtb UT127 or UT205, of which glycine, valine, alanine, asparagine, and leucine are nitrogen sources available to Mtb from human macrophages [72,73,74,75] (Appendix A). Although our metabolic modeling approach showed serine uptake, experimental results indicate that this amino acid is unavailable in sufficient quantities within a macrophage, but Mtb synthesized it [75].

The two strains were predicted to consume different amounts of nutrients, with UT127 taking up increased amounts of amino acids, such as D-Alanine, L-Serine, L-Isoleucine, Glycine, and L-Lysine, compared with Mtb UT205 (Appendix A). In concordance with the predicted higher flux through the TCA cycle, UT127 consumed pyruvate and fumarate (Figure 5B). Mtb UT205 consumed larger L-lactate quantities, ammonium, phosphate, glycerol, and L-proline (Appendix A).

Since glycolysis is an essential metabolic route triggered by the infection of macrophages with Mtb [32], we examined the expression of genes involved in glycolysis in the infected hSMs. In general, genes associated with glycolysis were relatively similar in hSMs infected with UT127 or UT205 (Appendix A). There was a coordinated upregulation of genes participating in the central macrophage metabolism. There was an upregulation of genes participating in glucose transport, mainly *GLUT6*, the glycolytic enzymes *HK1* and *HK2*, the phosphofructokinase *PFKFB3*, the enolases *ENO3*, and *ENO2*, and the lactate dehydrogenase *LDHA*, similar to the Warburg effect in cancer cells [32]. This profile is reminiscent of the transcriptional response of mouse bone-marrow-derived macrophages infected with the clinical strains of Mtb, CDC1551, a strain from the LAM family, and the hypervirulent strain of the Beijing family, HN878 [32]. Of interest, the 6-Phosphofructo-2-Kinase/Fructose-2,6-Biphosphatase 3 (*PFKFB3*) expression levels were higher in hSMs infected with Mtb UT127 compared with UT205. PFKFB3 controls glycolysis in eukaryotes by participating in the synthesis and degradation of fructose-2,6-bisphosphate. It has been shown that the expression of the mouse *pfkfb3* is differentially regulated in mouse bone marrow-derived macrophages upon infection with different Mtb strains. In this case, at 6 h of infection of BMDMs, the CDC1551 strain of Mtb expressed higher levels of *pfkfb3* compared with the hypervirulent HN878 strain of Mtb [32]. It is also apparent from our results that in addition to the expression of the *PFKFB3* gene, infection of hSMs with the UT127 strain also induced higher levels of *HIF1A*, *HK1*, *HK2*, *PDP1*, and *PDP2* genes compared with the infection with UT205 (Appendix A).

These results suggest an increased glycolytic flux in hSMs infected with UT127 compared with UT205, as well as a more marked Warburg effect, suggesting a more pronounced M1-like effect in hSMs infected with UT127 compared with UT205.

## 3. Discussion

As extra pulmonary survival of Mtb is a poorly studied but important stage of this pathogen’s life cycle, we developed a human splenic macrophage model for TB [34,36]. Using clinical strains of Mtb, we further applied dual RNA-seq to define this model system. Host–pathogen interactions are a delicate interplay. Previous work has highlighted how macrophage ontology manipulates Mtb metabolism and survival. Here, we explored how different clinical strains of Mtb manipulate the host macrophage environment. We show that our clinical strains induced a unique immune response despite high genetic similarities. Applying a macrophage–Mtb genome-scale model further allowed us to predict that the metabolic response to these strains was also distinct. This adds to a body of evidence showing that different strains of Mtb direct individual responses.

The UT205 Mtb drives a type I IFN, innate immunity, cell death, and granulocytes compared to UT127. Different studies in the mouse model involve type I IFNs and neutrophil recruitment with a higher inflammatory response and cell death [76], and neutrophil extracellular traps (NETs) were found in necrotic lung lesions of active TB patients [77]. Of interest, Mtb infection-induced pathology and cell death levels in mice also vary with the strain virulence [8,78,79]. Thus, our findings in human tissue macrophages infected with two closely related strains of Mtb of the LAM family are also concordant with an interpretation suggesting that UT205 displays a transcriptomic response associated with increased virulence, compared with UT127.

Our results on the metabolic reconstruction further complement the differences observed at the transcriptional level. For example, the results of the cumulative flux reactions indicate that the infection with UT205 induced a higher flux towards glycolysis/gluconeogenesis compared with UT127. Conversely, infection with UT127 resulted in a higher flux in the citric cycle and alanine metabolism and aspartate. Interestingly, alanine has been reported to be a byproduct of the metabolism of inflammatory macrophages [80]. It is well established that glycolysis plays a central role in the metabolism of M1 macrophages [81]. Moreover, aspartate is a common component of the aspartate–arginosuccinate shunt that participates in the citrulline and the citrulline–arginine–NO cycle, a central effector molecule of M1 macrophages [82]. Thus, both clinical isolates can induce an inflammatory response, but the molecular strategies seem to be distinct for each isolate.

Our integrated metabolic modeling of hSMs infected with Mtb shows a notorious higher number of reactions, including fatty acid oxidation, pyruvate metabolism, chondroitin synthesis, inositol phosphate metabolism, and alanine metabolism aspartate amino acids, in hSMs infected with UT127 compared with UT205. On the other hand, UT205 responded to the macrophage environment by increasing RH reactions associated with lipid metabolism and cholesterol degradation compared with UT127.

Although macrophage activation has been defined under different experimental settings [83], macrophage response’s M1/M2 paradigm under polarizing conditions has received remarkable attention. The M1 macrophages have been described as those responding to IFN-γ plus infection and a danger-associated molecular pattern (DAMPs). Conversely, M2 macrophages are influenced mainly by type-2 cytokines, IL-4. In this sense, macrophage ontogeny and micro-environmental signals shape the macrophage response to infection [84]. Furthermore, M2 macrophages are characterized by cholesterol homeostasis and lipid synthesis [81]. Thus, during early infection (6 h), no clear M1 or M2 polarized response is distinguished in hSMs infected with Mtb, prevailing an M1-like/M2-like phenotype in hSMs infected with UT205 or UT127, respectively. Although the M1/M2 paradigm seems to fit murine macrophages, recent data from human macrophages points to a more complex picture. It has been recently described that human alveolar macrophages from different human populations display a combined surface expression of M1 and M2 markers [85], and this hybrid phenotype may confer the ability of AMs for appropriate responses to stimuli and tissue environments [85]. This mixture of M1/M2 phenotypic markers has also been observed in human macrophages obtained from different pathological states [86,87,88]. Our results showed no clear distinction towards the M1 or M2 phenotype of hSMs infected with either clinical isolates of Mtb or infected macrophages, displaying genes associated with M1 and M2 phenotypes. As depicted in Figure 6, hSMs infected with the clinical isolates UT127 or UT205 responded to infection by expressing M1- and M2-associated markers. Infection with UT127 led to the expression of the M1 markers IL12RB, IL2RB, and IFNG, concomitantly expressing the M2 markers CTLA4, IL10, and PLXNB2 [83].

Furthermore, macrophages infected with UT127 showed downregulated oxidative phosphorylation and apoptosis and increased myeloid-derived suppressor cell category, M2-associated markers, while showing up-regulated glycolysis, angiogenesis, Th17 induction, and M1-associated markers [83]. Similarly, macrophages infected with UT205 displayed upregulation of the M1-associated markers IFNG, IL15, and CSF2RB and the M2 markers TGFB2 and IL10 [83], upregulation of biological processes associated with Type 1 IFNs, necroptosis, and glutamate amino acids (alanine, aspartate), and downregulation of processes such as apoptosis and oxidative phosphorylation. Of interest, the oxidative phosphorylation functional category was significantly downregulated in hSMs infected with both strains (Figure 1B). It was recently demonstrated that THP-1 and human MDMs infected with Mtb H37Rv decreased basal respiration and glycolysis to promote Mtb metabolic quiescence [89], reinforcing the conclusion that upon infection, Mtb rapidly alters the macrophage metabolism allowing for better intracellular survival.

In general, Mtb responded to the stressed conditions imposed by the intracellular macrophage environment by up-regulating pathways associated with cell-wall remodeling, hypoxia, fatty acids, and lipid metabolic processes, as well as genes associated with the immune response posed by the macrophage, among others. Interestingly, UT127 adapted to macrophage conditions by up-regulating genes mainly associated with iron acquisition pathways and controlling the metabolic interference of metal cations (copper, cadmium) pumped by the macrophage inside the phagosome. Meanwhile, UT205 mainly up-regulated genes associated with cholesterol and the lipid metabolic process. Although these biological processes have been described as essential responses of Mtb mainly through infection of animal models with selected mutants of the bacteria [90], no data are currently available on the actual early gene expression profile of circulating clinical strains of Mtb inside human tissue macrophages. Therefore, our data indicate that two related strains from the LAM family of Mtb with a highly similar genome identity [22] display distinct genetic adaptation programs to the infected macrophages’ particular stressful conditions.

This paper has described the transcriptomic response of human splenic macrophages to infection with two clinical isolates of Mtb’s LAM family. Although our results suggest a distinct pattern of response to each Mtb isolate, likely depending on each isolate’s transcriptomic differences, it may be possible that other macrophage populations, such as alveolar macrophages, respond differently to Mtb infection compared to splenic macrophages. Limited evidence suggests that this may be the case. For example, a metabolomic study of mouse lung and spleen infected with Mtb H37Rv demonstrated significant responses to the infection in several metabolites, including AMP, lactate, NADP+ succinate, and others, between the lung and spleen [91]. Further, mouse splenic and alveolar macrophages differ in their basal transcriptomic profiles [84,92]. Our group recently demonstrated that human splenic macrophages displayed an attenuated transcriptomic response to infection with the clinical isolates UT127 and UT205 compared to alveolar macrophages from healthy individuals and patients with pulmonary tuberculosis [28]. The recently described reports of Huang and colleagues [5] and Pisu and colleagues [7] elegantly demonstrate the differences in mouse alveolar and interstitial macrophages to control the infection with Mtb, suggesting a distinct response to Mtb infection depending on the particular macrophage population.

Finally, our results agree with the notion that the differential growth of Mtb in macrophage populations, such as hSMs, is determined by the metabolic changes induced by the bacteria and the adaptive metabolism of the bacteria in response to these changes. The macrophage metabolism is directly linked to the immune response, such as the case of type I IFNs correlating with the bacterial burden, early cholesterol utilization by Mtb, and the role of iron and fatty acids for Mtb colonization of macrophages [7,29]. Besides, genome-scale modeling, further constraint associated with transcriptomic data dramatically improves the in silico representation of a cell system’s metabolic capabilities and undoubtedly contributes to detailed physiological insights about complex metabolic responses during Mtb infection. Our observations limited to a one-time point (6 h) impede us from concluding the consequences of Mtb infection in later time points and defining whether a clear M1- or M2-like polarization state associates with UT205 or UT127 infection. In addition, the small sample number and the use of a limited number of Mtb clinical isolates are also limitations of our study. However, our study design used millions of tissue macrophages, most of which were infected with Mtb, and the power of the RNA-seq methodology allowed us to simultaneously examine the transcriptomic response of the macrophages and Mtb isolates. Therefore, we expect that our data overall may contribute to a deeper understanding of the macrophage–Mtb interaction.

## 4. Materials and Methods

### 4.1. Mycobacterium tuberculosis Clinical Strains

The Mtb clinical isolates UT205 and UT127 were obtained from the Centro Colombiano para la Investigación en Tuberculosis (CCITB). They were collected during a large cohort study conducted by the CCITB from 2005–2009 [93]. Both clinical isolates belong to the Latin American and Mediterranean family (LAM) of Mtb [94]. UT205 and UT127 used for infection were grown until the mid-log phase in Middlebrook 7H9 liquid medium (Difco, Sparks, MD, USA) supplemented with 10% of oleic acid–albumin–dextrose–catalase (OADC, BD BBL Middlebrook, Thermo Fisher Scientific, Waltham, MA, USA), 0.5% Glycerol (Promega, Madison, WI, USA), and 0.05% Tyloxapol (Sigma, Saint Louis, MO, USA). Batches were aliquoted in RPMI 1640 (Gibco, Thermo Fisher Scientific, Waltham, MA, USA) containing 30% glycerol and then frozen at −70 °C until used for infection. Axenic cultures of Mtb UT127 and UT205 were grown in Sauton’s medium [95].

### 4.2. Human Splenic Macrophages and Infection

As previously described, human splenic macrophages (hSMs) were obtained from spleen mononuclear cells from deceased patients [28,34,36]. Samples included six deceased donors (*n* = 6, Female: 2, Male: 4, mean age: 38 years) from the Transplantation Programs at the Hospital Universitario Pablo Tobón Uribe, and the IPS Universitaria León XIII Sede Medellín (Medellín, Colombia). Macrophages were seeded at 1 × 10^6^ viable cells per well in 6-well plates and cultured overnight in RPMI-1640 supplemented with 10% inactivated AB+ serum in the absence of antibiotics. Cells were washed with pre-warmed PBS containing 1% AB+ serum to eliminate cellular debris and then loaded with 1 mL of pre-warmed RPMI-1640 supplemented with 10% inactivated AB+ serum in the absence of antibiotics. Replica wells (5–10) were left uninfected or infected for 6 h (MOI 10:1) with Mtb cultured as described [22]. Briefly, macrophages were infected with Mtb UT127 or UT205 logarithmically grown (OD_600_) in 7H9. Since macrophage phagosomes have been described as a nutrient-limited environment for mycobacteria [96], logarithmic cultures of both strains of Mtb were grown for 6 h in Sauton’s minimal medium. Sauton’s minimal medium has been frequently used to test the growth conditions of Mtb that partially mimic some of the conditions that may be found in the mycobacterial-containing phagosome [97,98,99,100,101,102]. Finally, the transcriptomes of intracellular Mtb were compared with the transcriptomes of the strains cultured for 6 h in the Sauton’s medium. The infection of multiple replica wells compensates for variability in the proportion of adhered cells per well and the percentage of infected cells. In pilot experiments, Zeel–Nielssen staining showed that approximately 88% of the adherent cells were associated with a least one acid-fast bacilli (AFB) per infected macrophage (Appendix A). In summary, to estimate the % of infected macrophages and the AFB/infected macrophage, cells were stained by the Ziehl–Neelssen method. Initially, cells were stained with a pre-heated (50 °C) Fucsin solution for 3 min, rinsed with tap water to eliminate excess stain, decolorated with acid–alcohol, and then incubated with a solution of methylene blue for 20 s. To eliminate excess methylene blue, cells were washed with tap water and air-dried. To estimate the % of infected macrophages, at least 200 hundred cells were counted by selecting random fields (1000×) and counting cells with at least 1 AFB. To calculate the number of AFB/infected macrophages, the method described by [103] was used. Thus, the transcriptomes analyzed represented the infected macrophages rather than the transcriptome of bystander macrophages.

### 4.3. Dual RNA-Seq

A total of nine libraries derived from hSMs were prepared for RNA-seq sequencing. Samples included three biological replicates and three conditions (non-infected, infected with Mtb UT127, or UT205). Further, transcriptomes of Mtb UT127 and UT205 cultured under axenic conditions for 6 h in a carbon-poor medium (Sauton’s) were used to approximate a condition that may be confronted by Mtb inside macrophages [95]. To minimize the introduction of additional technical variability sources, all samples were stored at −80 °C in RLT buffer until RNA extraction and were processed in a single batch.

Total RNA from Mtb grown in Sauton’s minimal medium was obtained as described elsewhere [104]. Total RNA of Mtb-infecting hSMs was obtained using the RNeasy Plus Mini Kit (Qiagen) following the manufacturer’s recommendations. Briefly, cold RLT buffer supplemented with 1% β-mercaptoethanol was added to the monolayer at 6 h post-infection, and the cells were transferred to 50 mL Falcon tubes and subjected to two cycles of tissue homogenization for 2 min (previously frozen with liquid nitrogen) and 1 min incubation on ice between cycles. The homogenate was transferred to a 1.5-mL screw-cap tube containing zirconium-silica beads (Benchmark, Sayreville, NJ, USA), and mycobacteria were disrupted by bead beating using a FastPrep instrument (six cycles of 30 s at maximum speed with cooling on ice between cycles). All procedures were executed at a cold (4 °C) temperature to chase the macrophages’ RNA expression state. The sample was centrifuged for 10 min at 4 °C and 10,000 rpm, and the supernatant was passed through gDNA eliminator columns. According to the manufacturer’s instructions, total RNA was collected by RNA spin columns (RNeasy Plus Mini Kit, Qiagen, Hilden, Germany). The RNA’s purity was assessed by measuring the absorbance ratio at 260 and 280 nm using a Nanodrop 2000 Spectrometer (Thermo Scientific, Waltham, MA, USA). All samples displayed a 260/280 ratio of 2.0 or higher. The integrity of the RNA was assessed using an Agilent 2100 Bioanalyzer (Agilent Technologies, Santa Clara, CA, USA). Only samples with an RNA integrity number (RIN) > 8 were considered. The final number of RNA samples for RNA sequencing was 11: Six RNA samples of Mtb-infected hSMs, three RNA samples of non-infected hSMs, plus two RNA pooled samples of axenic cultures of UT127 and UT205 from three independent culture experiments.

RNA-Seq libraries were sequenced using an Illumina platform (Macrogen, Seoul, Korea). rRNA from macrophages and bacteria was depleted (Illumina Ribo-Zero Gold rRNA Removal Kit -Epidemiology, San Diego, CA, USA), and sequencing was performed (TrueSeq, stranded, paired-end reads of 100 bp) using a HiSeq2500 platform. Raw reads were trimmed to Q30 retaining the sequences with a minimum length of 70 bp. Analysis of transcriptomic data from infected SMs and Mtb isolates was performed using the mapping package STAR [105] v020201 and the statistics package edgeR [106] v3. Read pairs mapping was performed using the STAR default setting in the human genome hGR38 and Bowtie2 [107] in the *M. tuberculosis* H37Rv genome reference. Genome annotation for the human reference was obtained from the Ensembl website. Each of the six RNA-seq libraries from hSMs generated between 42.3 and 49.1 million reads pairs. We mapped the above quality thresholds, between 85.3% and 88.3% of the reads, to the human reference genome (Appendix A). Human-origin ribosomal reads were estimated to range between 2.4% and 16.4%. Raw counts for each gene were used for subsequent differential expression analysis with the R studio and EdgeR program following the program manual’s protocol.

In *M. tuberculosis*, the mapped reads ranged between 0.34% and 0.94%, of which between 1.6% and 3.8% were of bacterial ribosomal origin. With at least one read, we could detect the mycobacteria between 3699 and 3912 genome-annotated features (Appendix A). Raw counts for each genome feature were estimated with the HTseq program [108], with the HTseq-count function with the stranded option enabled.

### 4.4. Bioinformatics and Differential Gene Expression Analysis

The HTseq package was used to assign uniquely mapped reads to exons and genes using a species-specific gene annotation file from Ensembl to generate raw count data. Once raw count data were generated, data were filtered and normalized with the Trimmed Mean of M-values (TMM) method implemented in edgeR. We performed a principal component analysis (PCA) on the TMM-normalized data to explore the samples’ clustering per treatment group. For differential expression analysis, a gene feature read count was carried out with the HTseq program, and then a global table of reading counts (log CPM) per sample was organized. This table was used as the input gene counts for the edgeR analysis. EdgeR analysis results were printed as graphs and tables, and saved *p*-values were adjusted for multiple testing using the Benjamini and Hochberg correction [109]. A cutoff of less than 0.05 for *p*-values, 0.05 for the false discovery rate (FDR), and an absolute fold change (FC) value greater than 1.5 compared to media control were used to identify significant differentially expressed genes.

Mycobacterial transcriptomes were first upper quartile normalized. Highly expressed genes were selected on the normalized count values higher than percentile 90.

### 4.5. Enrichment Analysis

For infected hSMs, gene enrichment was done using the complete list of differentially expressed genes and examined with GO-seq [39]. Further analysis (Biological Process and KEGG pathways) focused on the immune response genes extracted from the ImmPORT database (The Immunology Database and Analysis Portal) that includes a comprehensive list of human immune-related genes [37]. The STRING database (v10.5) of known and predicted protein–protein interactions [110] was further used to identify genes with closely related functions and GO enrichment terms.

We performed a Gene Ontology (GO) Enrichment using the STRING database with default parameters [111]. This allowed inference of the biological processes and the protein–protein interaction network in which these genes are involved. We analyzed them using the signaling pathway analysis (SPIA) algorithm to confirm the results obtained from the previous analysis. This algorithm quantifies the perturbation that a given gene or group of genes affects a particular signaling pathway [40].

### 4.6. Host–Pathogen Genome-Scale Metabolic Network

An improved Mtb genome-scale model was used to integrate transcriptomics data from Mtb (sMtb2.0) [66]. The Mtb model comprised 989 genes, 1198 intracellular reactions, and 122 exchange reactions. This Mtb model was merged with the macrophage tissue metabolic network published by Zimmerman and colleagues [30], following the steps described by Jamshidi and Raghunathan [112]). The stoichiometric matrices from Mtb and macrophage were merged through the phagosome compartment, where Mtb and macrophage share common metabolites. Around 61 transport reactions allow the exchange of metabolites between the cytosol and phagosome compartment with Mtb. The metabolites exchanged between macrophages and Mtb were amino acids, fatty acids, nucleotides, metal ions, and sugars. The relationships between genes, proteins, and reactions (GPR, gene–protein reaction associations) were updated with a new Recon, Recon2.2 [113] version. Old NCBI identifiers were changed by actualized HGNC (HUGO Gene Nomenclature Committee). We developed an interdependence test to identify the coupling influence between host and pathogen models [66] (Appendix A). The combined macrophage–Mtb model encompassed 4773 reactions, 3631 metabolites, 2467 genes, and eight compartments, including the phagosome compartment.

### 4.7. Integration of Transcriptomics Data into Host–Pathogen Models of Mtb

To gain additional insights into Mtb clinical isolates’ distinct adaptations to the tissue macrophage environment, a host–pathogen metabolic reconstruction was based on the intracellular transcriptomes differentially expressed by the UT127 and UT205 clinical isolates. A constraint-based genome-scale metabolic model was used [114]. We used the alveolar macrophage metabolic model from the global human metabolic network’s reactions, Recon 1 [115]. Then, we used our Mtb genome-scale model, with the alveolar macrophage model version of Zimmerman and colleagues [30], to represent the human splenic macrophages (hSMs).

The Integrative Metabolic Analysis Tool (iMAT) [116] was implemented in Matlab with Gurobi 7.5.2 as an optimizer for gene expression data integration into the metabolic models. Here, all genes that passed the threshold of |log_2_ fold-change| > log_2_ 1.5, *p*-value < 0.05, and FDR < 0.05 were converted into tri-valued expression states. The flux activity state resulted from a Boolean mapping of gene–protein reaction associations (GPRs) in the model, identifying each reaction’s expression state based on those genes that were highly, moderately or lowly expressed (1, 0, and −1, respectively). The logical ‘AND’ and ‘OR’ operators were replaced with ‘max’ and ‘min’ expressions, respectively. The host–pathogens metabolic model’s solution involved developing a mixed-integer linear programming problem (MILP) to find a steady-state metabolic flux distribution to maximize the reaction activity’s consistency with their expression state satisfying stoichiometric and thermodynamic constraints (defined as the upper and lower bounds of each reaction). The optimization seeks to maximize the number of highly expressed active reactions (denoted as R_H) and the number of lowly expressed (denoted as R_L) inactive reactions.

## Figures and Tables

**Figure 1 ijms-23-01803-f001:**
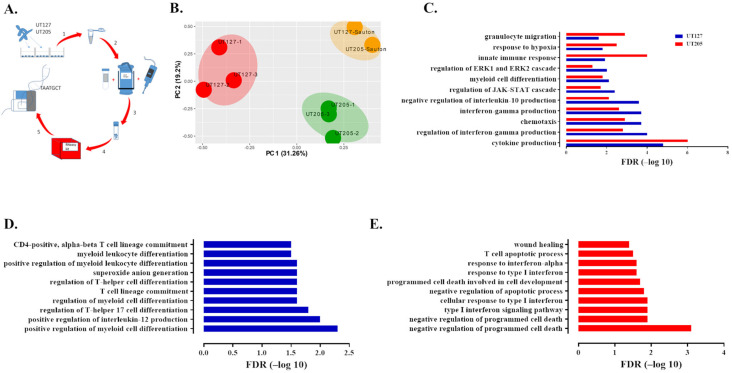
UT127 and UT205 clinical isolates of Mtb trigger a distinct transcriptomic response in human splenic macrophages upon infection. (**A**) General outline of the dual RNA-seq strategy. 5–20 × 10^6^ human splenic macrophages (hSMs) were infected for 6 h at an MOI of 10:1 with the two closely related clinical strains of the Latin American and Mediterranean (LAM) family of Mtb, UT127, and UT205. Upon total RNA-recovery, samples were RNA-seq sequenced and reads mapped to the human and Mtb genomes (see Section 4). (**B**) Principal component analysis (PCA) of Mtb transcriptome cluster gene expression of UT127 from UT205 inside infected hSMs. Differentially expressed genes for UT127 and UT205 were obtained by comparing the same strains cultured in a carbon-poor medium (Sauton’s). (**C**–**E**) Deduced functional categories using the GO-seq algorithm on the immune genes obtained from the ImmPORT database. (see Section 4). Infection of hSMs with UT127 was associated with 645 over-represented biological processes, while infection with UT205 was associated with 585 over-represented processes. (**C**) Illustrates selected common biological processes differentiating the hSM response to UT127 and UT205. (**D**,**E**) Selected unique immune biological processes over-represented in response to UT127 (**D**) or UT205 infection (**E**). Blue and red bars represent biological processes (**C**–**E**) associated with the hSMs transcriptional response to Mtb UT127 or UT205.

**Figure 2 ijms-23-01803-f002:**
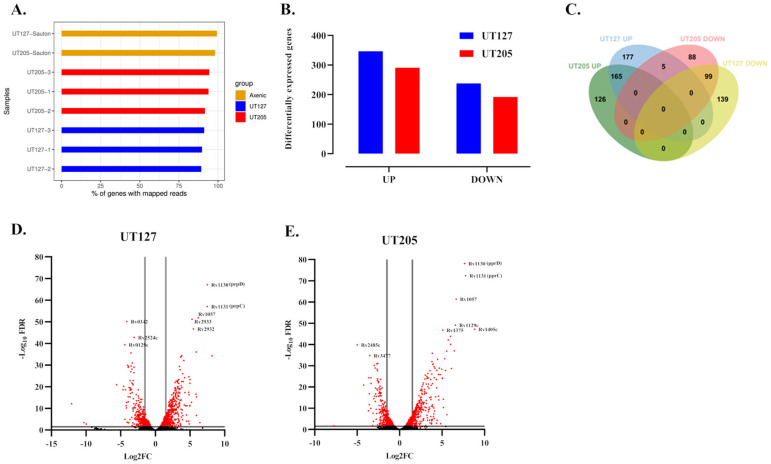
Gene expression response of UT127 and UT205 Mtb strains to hSMs adapt to hSMs by expressing distinct transcriptomic profiles. (**A**) Bar graph displaying the percentage of genome coverage of the reads obtained from intracellular Mtb. Reads from intracellular Mtb UT127 and UT205 of each triplicate sample were mapped to the Mtb H37Rv genome (see Section 4). Reads mapped to 80–90% of the Mtb genome, depending on the sample. (**B**) Bar chart indicating the amount of differentially expressed genes (DEGs) for each condition. (**C**) Venn diagram displaying the sharing of differentially expressed up-regulated and downregulated genes of Mtb UT127 and UT205 in infected hSMs. (**D**,**E**) Volcano plots of the genes expressed by Mtb strain UT127 (**D**) and UT205 (**E**) within 6 h of infection of human splenic macrophages. Macrophages were infected for 6 h with an MOI of 10:1. Differentially expressed genes were selected with 1.5 Log_2_FC, *p* < 0.05, FDR < 0.05. Grey lines discriminate differentially expressed genes (DEGs). The most highly up-regulated and downregulated genes are shown.

**Figure 3 ijms-23-01803-f003:**
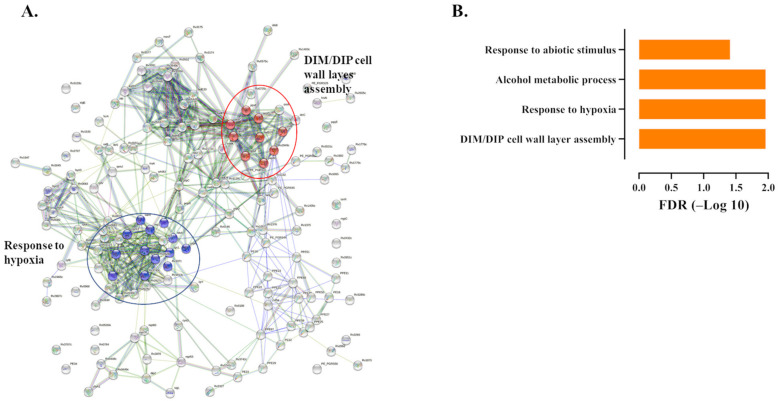
Protein-Protein interaction network of commonly up-regulated genes expressed by Mtb within 6 h of hSMs infection. (**A**) Protein-Protein interaction network of commonly up-regulated differentially expressed genes (*n* = 165) as deduced from the String database. The 2 top GO categories, DIM/DIP cell wall layer assembly (FDR 0.00031) and response to hypoxia (FDR 0.00031), are encircled. (**B**) Significant biological processes (FDR < 0.05) associated with the 165 commonly up-regulated differentially expressed genes by hSMs infected for 6 h with M. tuberculosis.

**Figure 4 ijms-23-01803-f004:**
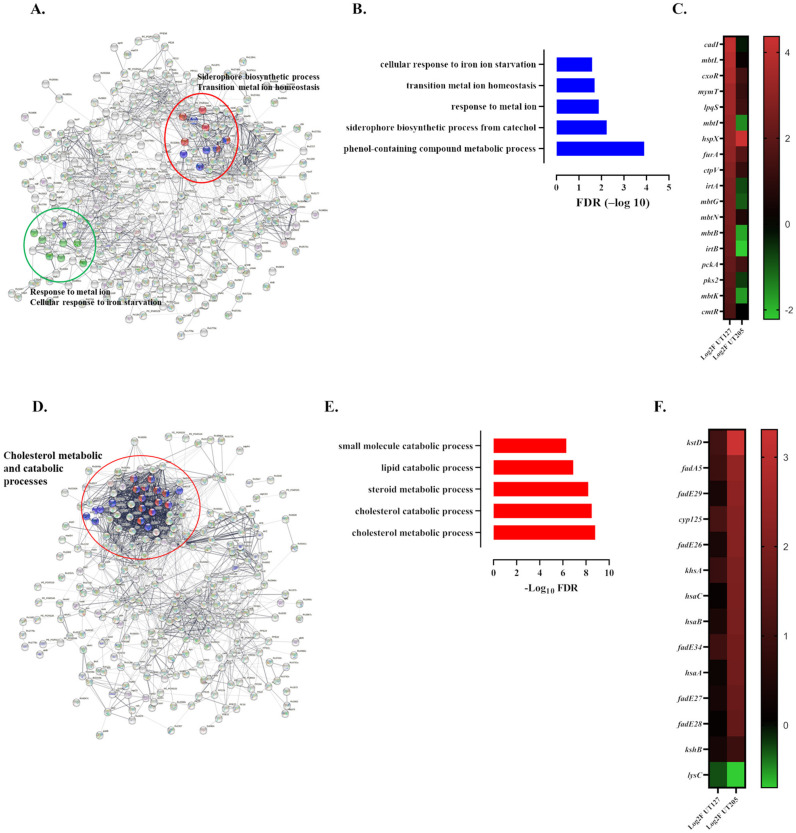
UT127 and UT205 Mtb strains adapt to hSMs by expressing distinct transcriptomic profiles. (**A**,**D**) Protein-Protein interaction networks associated with the up-regulated DEGs of UT127 (*n* = 182) (**A**) and UT205 (*n* = 126) (**D**) as determined by the String database. Encircled clusters identify genes associated with selected biological processes. (**A**,**D**). Top significant biological processes (FDR < 0.05) based on Gene Ontology (GO) of up-regulated genes of UT127 ((**B**), blue bars) and UT205 ((**E**), red bars) within 6 h of infection of hSMs. (**C**,**F**) Heat maps representing the expression (Log_2_ Fold) and significance (FDR, False Discovery Rate, right) of the genes associated with the GO categories, siderophore biosynthetic process, response to metal ion, transition metal homeostasis, and cellular response to iron starvation of the UT127 strain, compared to the UT205 strain (**C**), and the GO categories, alcohol metabolic process, cholesterol metabolic process, steroid metabolic process, and cholesterol metabolic process of the UT205 strain, compared with the UT127 strain (**F**).

**Figure 5 ijms-23-01803-f005:**
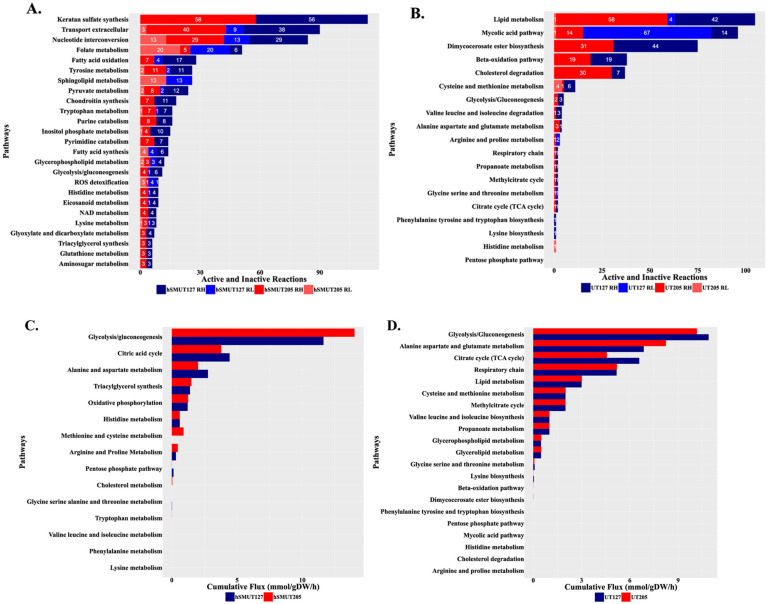
A transcriptomic-based reconstruction of the host–pathogen early interaction indicates a distinct interaction. (**A**,**B**) Highly expressed and lowly expressed reactions in the Host–Pathogen Metabolic Network with Mtb UT127 and UT205 as a pathogen. (**A**) Number of highly expressed (RH) and lowly expressed (RL) reactions in UT127- and UT205- infected human spleen macrophages metabolic pathways. (**B**) Number of highly expressed (RH) and lowly expressed (RL) reactions in Mtb UT127 and UT205 metabolic pathways. Numbers inside the colored squares indicate the number of reactions. (**C**,**D**) Cumulative fluxes derived from the integration of transcriptomics data into the Host–Pathogen Metabolic Network. (**A**) Cumulative flux by Mtb-infected hSMs metabolic pathways; (**B**). Cumulative flux by Mtb UT127 and UT205 metabolic pathways.

**Figure 6 ijms-23-01803-f006:**
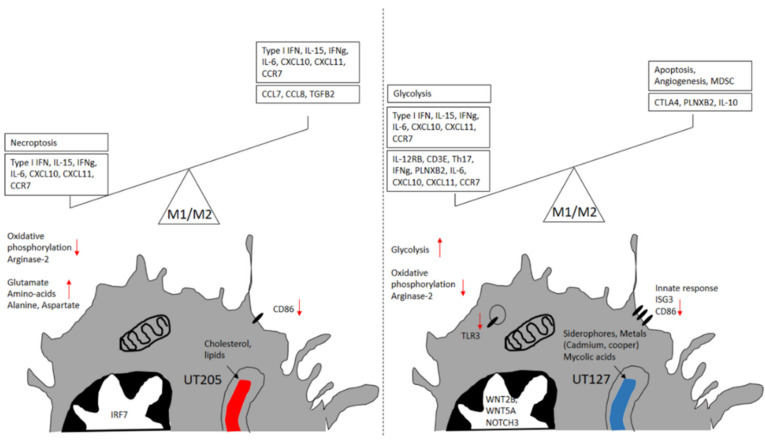
Splenic macrophages infected with *M. tuberculosis* UT127 and UT205 display M1- and M2-like characteristics. Within 6 h of infection with Mtb, hSMs express a mixture of M1 and M2 polarization markers. Red arrows represent genes or cellular processes associated with the M1 macrophage polarization. Red arrows pointing down represent genes or cellular processes associated with the M2 macrophage polarization. There is no clear association with M1 or M2 polarization in hSMs infected with Mtb UT127 or UT205. The direction of the upward and downward arrows indicate upregulated or downregulated genes, respectively.

## Data Availability

Data were submitted to the Sequence Read Archive (SRA) under BioProject ID PRJNA778012.

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
