# Peer review of "Dual RNA Sequencing of Mycobacterium tuberculosis-Infected Human Splenic Macrophages Reveals a Strain-Dependent Host–Pathogen Response to Infection"

_ijms, 2022, doi:10.3390/ijms23031803_

Round 1
Reviewer 1 Report
The manuscript “Dual RNA Sequencing of Mycobacterium tuberculosis-infected Human Splenic Macrophages Reveals Strain-dependent Host-pathogen Response to Infection” by Víctor A. López-Agudelo and colleagues, the authors used dual RNA-seq, to profile the transcriptomic cross-talk between human splenic macrophages infected with two clinical isolates of the Latin American and Mediterranean family of Mtb (UT127 and UT-205). The authors showed that, there was a more robust innate immune response to UT127 clinical isolate, suggesting an early and more potent recognition by the innate immune system than the UT205 strain, which may have consequences in the progression of the disease and found a mixture of M1/M2-like polarization in the splenic macrophages infected with the clinical isolates.
The study was well conducted, but I have one major concern regarding the controls used.
The authors compared Mtb cultured in Middlebrook 7H9 and infected in splenic macrophages to Mtb cultured in Sauton´s minimal medium. If the question of this manuscript is to study the gene expression of Mtb inside splenic macrophages, then the culture medium used to grow Mtb should be the same in both conditions. Different culture media will induce different gene expression patterns in Mtb, especially at the metabolic level, leading to different metabolic pathways being activated in function of the carbon source available. This is particularly important because the authors draw conclusions about metabolic pathways induced by the Mtb strains. Therefore, this study should compare the gene expression of splenic macrophages to Mtb cultured in the same culture medium.
Minor concern: given that the authors are studying gene expression in two different strains of Mtb, would be relevant to know if the burden of infection is not conditioning the global gene expression. Therefore, to know the precise rate of infection for each strain, and to quantify the number of intracellular mycobacteria per macrophage would be very informative. Please show data on the rate of infection and number of intracellular mycobacteria per macrophage for each strain in splenic macrophages at 6h post-infection.
Author Response
The authors compared Mtb cultured in Middlebrook 7H9 and infected in splenic macrophages to Mtb cultured in Sauton's minimal medium. If the question of this manuscript is to study the gene expression of Mtb inside splenic macrophages, then the culture medium used to grow Mtb should be the same in both conditions. Different culture media will induce different gene expression patterns in Mtb, especially at the metabolic level, leading to different metabolic pathways being activated in the function of the carbon source available. This is particularly important because the authors draw conclusions about metabolic pathways induced by the Mtb strains. Therefore, this study should compare the gene expression of splenic macrophages to Mtb cultured in the same culture medium.
Answer: We thank the Reviewer for the comments and agree that culturing Mtb in the same medium would be more comparable. However, as previous studies used 7H9 grown Mtb for macrophage infections we wanted to recapitulate this to enable cross-experimental comparisons. However as the intracellular environment is considered to be nutrient-poor (Berney and Berney-Meyer, 2017), we wanted to use growth in minimal media as a control. Sauton's minimal medium is phosphate limited and has been used in other studies to mimic some of the conditions that may be found in the mycobacterial-containing phagosome (Piddington et al., 2000; Mariani et al., 2000; Shleeva et al., 2002; Florio et al., 2006; Sakthi and Narayanan, 2013; Goodsmith et al., 2015). A modified description of the 4.2. Human splenic macrophages and infection section of Materials and Methods, is now present in the resubmitted manuscript (lines 601-614).
References
- Berney, M.; Berney-Meyer, L., Mycobacterium tuberculosis in the Face of Host-Imposed Nutrient Limitation. Microbiol Spectr 2017, 5, (3).
- Florio, W.; Batoni, G.; Esin, S.; Bottai, D.; Maisetta, G.; Favilli, F.; Brancatisano, F. L.; Campa, M., Influence of culture medium on the resistance and response of Mycobacterium bovis BCG to reactive nitrogen intermediates. Microbes Infect 2006, 8, (2), 434-41.
- Goodsmith, N.; Guo, X. V.; Vandal, O. H.; Vaubourgeix, J.; Wang, R.; Botella, H.; Song, S.; Bhatt, K.; Liba, A.; Salgame, P.; Schnappinger, D.; Ehrt, S., Disruption of an M. tuberculosis membrane protein causes a magnesium-dependent cell division defect and failure to persist in mice. PLoS Pathog 2015, 11, (2), e1004645.
- Mariani, F.; Cappelli, G.; Riccardi, G.; Colizzi, V., Mycobacterium tuberculosis H37Rv comparative gene-expression analysis in synthetic medium and human macrophage. Gene 2000, 253, (2), 281-91.
- Piddington, D. L.; Kashkouli, A.; Buchmeier, N. A., Growth of Mycobacterium tuberculosis in a defined medium is very restricted by acid pH and Mg(2+) levels. Infect Immun 2000, 68, (8), 4518-22.
- Sakthi, S.; Narayanan, S., The lpqS knockout mutant of Mycobacterium tuberculosis is attenuated in macrophages. Microbiol Res 2013, 168, (7), 407-14.
- Shleeva, M. O.; Bagramyan, K.; Telkov, M. V.; Mukamolova, G. V.; Young, M.; Kell, D. B.; Kaprelyants, A. S., Formation and resuscitation of "non-culturable" cells of Rhodococcus rhodochrous and Mycobacterium tuberculosis in prolonged stationary phase. Microbiology (Reading) 2002, 148, (Pt 5), 1581-1591.
Minor concern: given that the authors are studying gene expression in two different strains of Mtb, would be relevant to know if the burden of infection is not conditioning the global gene expression. Therefore, to know the precise rate of infection for each strain, and to quantify the number of intracellular mycobacteria per macrophage would be very informative. Please show data on the rate of infection and number of intracellular mycobacteria per macrophage for each strain in splenic macrophages at 6h post-infection.
Answer: We used a MOI of 10:1 and infected for 6 hoursAt this MOI we determined that 85% of the splenetic macrophages were infected and that there was no significant difference between the two strains in terms of the AFB/infected macrophage. AFBs were calculated using the geometric mean as described in Crowle et al (1981), showing a median value of 6.3 for UT127, and 6.9 for UT205. The categories selected for this calculation were 1-5, 6-10, 11-20, and >20 AFB/infected bacilli.
A.
B.
Number of Acid-Fast Bacilli (AFB) per infected macrophage in human splenic macrophages infected with M. tuberculosis UT127 and UT205. 3 x 105 human splenic macrophages (hSMs) were infected for 6 hours with an MOI of 10:1 of either M. tuberculosis (Mtb) UT127 or UT205. Briefly, macrophages were seeded in 24-well plates for 24 hours, washed with a warm (37 oC) antibiotic-free medium to eliminate non-adherent cells, resuspended in RPMI-1640 supplemented with 10% AB+ medium in the absence of antibiotics. Then, approximately 3 x 106 CFUs of Mtb were added, and the plates were centrifuged 700 x g for 1 minute to increase the efficiency of Mtb: Macrophage contact and cultured for 6 additional hours. At the end of the phagocytosis period, the wells were extensively washed with warm (37 oC) antibiotic-free PBS to eliminate extracellular bacteria. Cells were fixed with 0.25% glutaraldehyde for 10 minutes at 37 oC, and then air-dried. To estimate the % of infected macrophages and the AFB/infected macrophage, cells were stained by the Ziehl-Neelssen method. Initially, cells were stained with a pre-heated (50 oC) Fucsin solution for 3 minutes, rinsed with tap water to eliminate the excess of stain, decolorated with acid-alcohol, then incubated with a solution of methylene blue for 20 seconds. To eliminate the excess of methylene blue, cells were washed with tap water and air-dried. To estimate the % of infected macrophages, at least 200 hundred cells were counted by selecting random fields (1000X) and counting cells with at least 1 AFB. To calculate the number of AFB/infected macrophages, the method described by (Crowle and May, 1981) was used. The cells were classified as containing no AFB, 1-5 AFB, 6-10 AFB, 11-20 AFB, and >20 AFB, and the geometric mean was calculated by 2 independent observers (A). Data were obtained from 3 independent experiments with the following results: UT127: 11.4, 6.9, and 11.7 AFB/infected macrophage; UT205: 9.4, 6.3, and 13.8 AFB/infected macrophage (Mann-Whitney P-value=0.9). In all cases, the % of infected macrophages was around 90% (B)
References
- Crowle, A. J.; May, M., Preliminary demonstration of human tuberculoimmunity in vitro. Infect Immun 1981, 31, (1), 453-64.
Reviewer 2 Report
The manuscript by Lopez-Agudelo and colleagues extends our views on M. tuberculosis-macrophages interplay during infection. The authors used human splenic macrophages as an extrapulmonary infection model, that is advantage, as extrapulmonary macrophages had been examined less comparing to traditional pulmonary macrophages. Infection was performed by two Colombia strains of LAM family, which had been characterized by these authors earlier in details. The strains are very close genetically, more than 99% of similarity.
The infected splenic macrophages were used for dual RNA-seq, that provided the authors with information of both host and pathogen transcriptomes at the very beginning of extrapulmonary infection (6hr). The raw sequencing data were comprehensively analyzed to obtain a big pool of transcriptome and metabolic differences between the two strains under comparison. The analysis showed that the strains with high genome similarity launch rather different response to infection.
The manuscript was written very carefully, with a lot of experimental details. The conclusions are informative and supported by the data. However, some clarification and explanation should be done:
- Why were the bacterial RNA-seq reads mapped on H37Rv genome? The genomes of both strains were sequenced and published by the authors earlier (Baena et al, 2019) and could be used as reference genomes.
- The strains used for infection were grown in rich Middlebrook 7H9 medium, but the control strains were cultured in poor-carbon Sauton medium. Why did Sauton medium was used? Earlier (Baena et al, 2019) the authors demonstrated that the compared strains had significant growth differences in Sauton, but grew similarly in Middlebrook. Could these differences in preliminary culturing of bacteria affect the transcriptome profiles?
- The discussion is well performed and very convincing. But I would like the authors to discuss the possible impact of genetic differences between the strains on their transcriptome profile in macrophages. In the previous paper the authors declared that they could not find any correlation between differences of bacterial transcriptomes in culture and their genomes. Perhaps, there is a correlation between genomes and transcriptomes in infected macrophages.
- There is a discrepancy in up-regulation of mycobactin siderophore synthesis: it was found to be upregulated in UT127 strain in macrophages (lines 297-298), but in UT205 in Sauton medium (Baena et al, 2019, page 701). Is there any explanation for this fact?

Author Response
- Why were the bacterial RNA-seq reads mapped on H37Rv genome? The genomes of both strains were sequenced and published by the authors earlier (Baena et al, 2019) and could be used as reference genomes.
Answer: The laboratory strain of Mtb, Mycobacterium tuberculosis H37Rv, has become the "reference" strain and in order to compare with their studies we, therefore, mapped to H37Rv.
- The strains used for infection were grown in rich Middlebrook 7H9 medium, but the control strains were cultured in poor-carbon Sauton medium. Why did the Sauton medium was used? Earlier (Baena et al., 2019), the authors demonstrated that the compared strains had significant growth differences in Sauton but grew similarly in Middlebrook. Could these differences in the preliminary culturing of bacteria affect the transcriptome profiles?
Answer: We used a MOI of 10:1 and infected for 6 hours. At this MOI we determined that 85% of the splenetic macrophages were infected and that there was no significant difference between the two strains in terms of the AFB/infected macrophage. AFBs were calculated using the geometric mean as described in Crowle et al (1981), showing a median value of 6.3 for UT127, and 6.9 for UT205. The categories selected for this calculation were 1-5, 6-10, 11-20, and >20 AFB/infected bacilli.
A.
B.
Number of Acid-Fast Bacilli (AFB) per infected macrophage in human splenic macrophages infected with M. tuberculosis UT127 and UT205. 3 x 105 human splenic macrophages (hSMs) were infected for 6 hours with an MOI of 10:1 of either M. tuberculosis (Mtb) UT127 or UT205. Briefly, macrophages were seeded in 24-well plates for 24 hours, washed with a warm (37 oC) antibiotic-free medium to eliminate non-adherent cells, resuspended in RPMI-1640 supplemented with 10% AB+ medium in the absence of antibiotics. Then, approximately 3 x 106 CFUs of Mtb were added, and the plates were centrifuged 700 x g for 1 minute to increase the efficiency of Mtb: Macrophage contact and cultured for 6 additional hours. At the end of the phagocytosis period, the wells were extensively washed with warm (37 oC) antibiotic-free PBS to eliminate extracellular bacteria. Cells were fixed with 0.25% glutaraldehyde for 10 minutes at 37 oC, and then air-dried. To estimate the % of infected macrophages and the AFB/infected macrophage, cells were stained by the Ziehl-Neelssen method. Initially, cells were stained with a pre-heated (50 oC) Fucsin solution for 3 minutes, rinsed with tap water to eliminate the excess of stain, decolorated with acid-alcohol, then incubated with a solution of methylene blue for 20 seconds. To eliminate the excess of methylene blue, cells were washed with tap water and air-dried. To estimate the % of infected macrophages, at least 200 hundred cells were counted by selecting random fields (1000X) and counting cells with at least 1 AFB. To calculate the number of AFB/infected macrophages, the method described by (Crowle and May, 1981) was used. The cells were classified as containing no AFB, 1-5 AFB, 6-10 AFB, 11-20 AFB, and >20 AFB, and the geometric mean was calculated by 2 independent observers (A). Data were obtained from 3 independent experiments with the following results: UT127: 11.4, 6.9, and 11.7 AFB/infected macrophage; UT205: 9.4, 6.3, and 13.8 AFB/infected macrophage (Mann-Whitney P-value=0.9). In all cases, the % of infected macrophages was around 90% (B)
References
- Crowle, A. J.; May, M., Preliminary demonstration of human tuberculoimmunity in vitro. Infect Immun 1981, 31, (1), 453-64.
- The discussion is well performed and very convincing. But I would like the authors to discuss the possible impact of genetic differences between the strains on their transcriptome profile in macrophages. In the previous paper the authors declared that they could not find any correlation between differences of bacterial transcriptomes in culture and their genomes. Perhaps, there is a correlation between genomes and transcriptomes in infected macrophages.
Answer: Differences in the transcriptomic profiles of Mtb UT127 and UT205 grown in axenic media (7H9 and Sauton's minimal medium) were presented and discussed by us in the paper published in Virulence (Baena et al., 2019). Even though they share a similarity higher than 99.6%, their transcriptomic profiles show distinct responses to the axenic media, particularly in the carbon-restricted medium. In this case, the transcriptome of UT205 grown in Sauton's medium showed that the predominant pathways for this clinical isolate are mainly associated with virulence; in contrast, UT127 showed pathways mainly associated with growth and survival. The results of our transcriptional studies sustain the idea that these two Mtb closely related circulating strains differ in their virulence programs. The results of our submitted manuscript, using human tissue macrophages, sustain and reinforce the conclusion that two closely related strains of Mtb can distinctly adapt their transcriptomic response to the restrictions imposed by macrophages.
References:
Baena, A.; Cabarcas, F.; Alvarez-Eraso, K. L. F.; Isaza, J. P.; Alzate, J. F.; Barrera, L. F., Differential determinants of virulence in two Mycobacterium tuberculosis Colombian clinical isolates of the LAM09 family. Virulence 2019, 10, (1), 695-710

Reviewer 3 Report
López-Agudelo and collaborators showed through dual-RNA seq and metabolic reconstructing that two closely related strains triggered different immune and metabolic responses in macrophages at the first six hours of infection. Moreover, they demonstrated that the strains exhibited a distinguished metabolic response to macrophage possibly indicating that mycobacteria metabolism influences the macrophage response. One of the great qualities of the study is the use of primary macrophages as a model and also the use of two closely related clinical strains. The content is of high significance for the researchers that investigate mycobacteria by giving relevant information about the immunometabolism of macrophages to different Mtb strains. However, the authors should be more careful with their figures. Many of the figures are very hard to see clearly due to the size and to the resolution. Please, see below additional comments/questions.
Major concerns:
1. The main concern is about the quality of the figures. The resolution of the figures is very low. The figures are also too small. The authors need to make larger figures with a better resolution (at least 300dpi).
2. How similar (% similarity) are the strains UT127 and UT205 in their genome?
3. The authors tested whether is there a difference in the capacity of both strains to survive to macrophages? UT205 apparently triggered a less inflammatory/responsive phenotype in macrophages (e.g., M2-phenotype and higher production of type I interferon). Does this reflect in a higher capacity to survive to macrophages? Is it known whether UT205 has a higher capacity to infect humans than the UT127 strain?
Author Response
- The main concern is about the quality of the figures. The resolution of the figures is very low. The figures are also too small. The authors need to make larger figures with a better resolution (at least 300dpi).
Answer: A new file containing the main figures at a higher resolution was sent to the Editors
- How similar (% similarity) are the strains UT127 and UT205 in their genome?
Answer: Higher than 99.6%, at the nucleotide level.
- The authors tested whether is there a difference in the capacity of both strains to survive to macrophages? UT205 apparently triggered a less inflammatory/responsive phenotype in macrophages (e.g., M2-phenotype and higher production of type I interferon). Does this reflect in a higher capacity to survive to macrophages? Is it known whether UT205 has a higher capacity to infect humans than the UT127 strain?
The authors tested whether is there a difference in the capacity of both strains to survive to macrophages?
Answer: We only tested the early response to Mtb infection at 6 hours before any significant replication would have occurred. Further experiments at later time points would be required and would be interesting but is beyond the scope of this study.
Is it known whether UT205 has a higher capacity to infect humans than the UT127 strain?
Answer: We used a MOI of 10:1 and infected for 6 hoursAt this MOI we determined that 85% of the splenetic macrophages were infected and that there was no significant difference between the two strains in terms of the AFB/infected macrophage. AFBs were calculated using the geometric mean as described in Crowle et al (1981), showing a median value of 6.3 for UT127, and 6.9 for UT205. The categories selected for this calculation were 1-5, 6-10, 11-20, and >20 AFB/infected bacilli.
A.
B.
Number of Acid-Fast Bacilli (AFB) per infected macrophage in human splenic macrophages infected with M. tuberculosis UT127 and UT205. 3 x 105 human splenic macrophages (hSMs) were infected for 6 hours with an MOI of 10:1 of either M. tuberculosis (Mtb) UT127 or UT205. Briefly, macrophages were seeded in 24-well plates for 24 hours, washed with a warm (37 oC) antibiotic-free medium to eliminate non-adherent cells, resuspended in RPMI-1640 supplemented with 10% AB+ medium in the absence of antibiotics. Then, approximately 3 x 106 CFUs of Mtb were added, and the plates were centrifuged 700 x g for 1 minute to increase the efficiency of Mtb: Macrophage contact and cultured for 6 additional hours. At the end of the phagocytosis period, the wells were extensively washed with warm (37 oC) antibiotic-free PBS to eliminate extracellular bacteria. Cells were fixed with 0.25% glutaraldehyde for 10 minutes at 37 oC, and then air-dried. To estimate the % of infected macrophages and the AFB/infected macrophage, cells were stained by the Ziehl-Neelssen method. Initially, cells were stained with a pre-heated (50 oC) Fucsin solution for 3 minutes, rinsed with tap water to eliminate the excess of stain, decolorated with acid-alcohol, then incubated with a solution of methylene blue for 20 seconds. To eliminate the excess of methylene blue, cells were washed with tap water and air-dried. To estimate the % of infected macrophages, at least 200 hundred cells were counted by selecting random fields (1000X) and counting cells with at least 1 AFB. To calculate the number of AFB/infected macrophages, the method described by (Crowle and May, 1981) was used. The cells were classified as containing no AFB, 1-5 AFB, 6-10 AFB, 11-20 AFB, and >20 AFB, and the geometric mean was calculated by 2 independent observers (A). Data were obtained from 3 independent experiments with the following results: UT127: 11.4, 6.9, and 11.7 AFB/infected macrophage; UT205: 9.4, 6.3, and 13.8 AFB/infected macrophage (Mann-Whitney P-value=0.9). In all cases, the % of infected macrophages was around 90% (B)
References
- Crowle, A. J.; May, M., Preliminary demonstration of human tuberculoimmunity in vitro. Infect Immun 1981, 31, (1), 453-64.
UT205 apparently triggered a less inflammatory/responsive phenotype in macrophages (e.g., M2-phenotype and higher production of type I interferon). Does this reflect in a higher capacity to survive to macrophages?
Answer: Concerning the M1/M2 polarized phenotypes, as shown in Figure 6 of the submitted manuscript, is that at 6 hours of infection, no clear M1or M2 polarization is observed. As discussed in the manuscript, the polarization phenotype as described in mice is controversial when human macrophages are infected in vitro with Mtb. In addition, the in vivo polarizing conditions are usually observed in mice weeks after infection, so it may be difficult to observe in a short period of time.
Finally, our transcriptomic analysis suggests a distinct Mtb strain-driven gene expression response of the macrophage, an observation previously shown by us in alveolar macrophages and monocytes from tuberculosis patients using microarray analysis (Lavalett et al., 2020a,b). In the submitted manuscript, the enrichment of upregulated genes of macrophages infected with UT205 significantly shows a predominant type I IFN response (Figure 1E), a higher granulocyte migration response (Figure 1C), and necroptosis (Supplementary Figure 3), suggesting that UT205 may be more virulent than UT127.
References
- Crowle, A. J.; May, M., Preliminary demonstration of human tuberculoimmunity in vitro. Infect Immun 1981, 31, (1), 453-64.
- Lavalett, L.; Ortega, H.; Barrera, L. F., Human Alveolar and Splenic Macrophage Populations Display a Distinct Transcriptomic Response to Infection With Mycobacterium tuberculosis. Front Immunol 2020, 11, 630

Round 2
Reviewer 1 Report
1) Please include the reference Crowle et al (1981), and the information provided in the answer to the second question (MOI and AFB calculations) in the material and methods section.
2) In the answer to question 2, the authors referred to Crowle et al (1981). In this paper, the authors claim, “For macrophages of most donors, usually between 20 and 40% of the cells became infected, most of these with between one and five acid-fast bacilli (AFB).” In the current manuscript, the authors refer that 85% of the macrophages were infected. This is substantially different from what was obtained by Crowle et al, as well as, by many other papers published in the field, in recent years. Please show evidence of 85% rate of infection in your samples.
Author Response
1) Please include the reference Crowle et al (1981), and the information provided in the answer to the second question (MOI and AFB calculations) in the material and methods section.
Answer: We appreciate the recommendations of the Reviewer. We have now included in the Materials and Methods section (lines 601-622) the corresponding description, as well as a new Supplementary Figure (Figure S8), describing the AFB/infected macrophage results.
2) In the answer to question 2, the authors referred to Crowle et al (1981). In this paper, the authors claim, "For macrophages of most donors, usually between 20 and 40% of the cells became infected, most of these with between one and five acid-fast bacilli (AFB)." In the current manuscript, the authors refer that 85% of the macrophages were infected. This is substantially different from what was obtained by Crowle et al, as well as, by many other papers published in the field, in recent years. Please show evidence of 85% rate of infection in your samples.
Answer: From the Crowle and May paper (1981), we used the method to calculate the AFB/infected macrophage. As described in this paper, the infected monocyte-derived macrophages (MDMs) for 30 minutes, and an MOI of 1-5. In our case, we infected tissue macrophages for 6 hours instead of 30 minutes, and as described in the legend of the figure we provided, plates were centrifuged 700 x g for 1 minute to increase the efficiency of Mtb: Macrophage contact and cultured for 6 additional hours, using an MOI of 10:1. These and other differences between Crowle’s paper and our experimental conditions can explain our higher increase in the number of infected cells.

Reviewer 2 Report
I didn't get answers that I could be satisfied with.
1. No explanation was given. I still do not understand why the genomic sequences of the strains under study were not used as references.
2. My question regarding the preparation of Mtb cultures for infection and the control strains remains open. In their response, the authors presented experimental data confirming the same efficacy of infection for both strains (AFB/infected macrophages data). But my concern was this: because of the different growth conditions (Sauton medium for the control strains and 7H9 medium for the strains used for infection), the strains acquired different metabolic characteristics BEFORE infection. That could influence the list of DEGs dramatically.
No answer for my point 4 was given.
Author Response
- No explanation was given. I still do not understand why the genomic sequences of the strains under study were not used as references.
Answer: We selected the genome of M. tuberculosis H37Rv as a reference since this is the genome with the best gene annotation so far. This genome has no gaps and has been extensively curated. On the other hand, the genomes of UT205 and UT127 were assembled into contigs, having unresolved repetitive genome elements. Additionally, the genomes of the three strains (H37Rv, UT205, and UT127) are highly conserved, with an average nucleotide identity of 99.9% among them. With such a high degree of nucleotide identity, the mapping software, Bowtie2, will have no problem assigning, in a precise manner, the RNA-seq reads to its respective genes.
A similar strategy, mapping WGS reads of clinical isolates to H37Rv genome reference, has been commonly applied in other research works (Welzen et al., 2017; Gomez-Gonzalez et al., 2019; Tang et al., 2020; Gupta and Alland, 2021)
- Cuevas-Cordoba, B.; Fresno, C.; Haase-Hernandez, J. I.; Barbosa-Amezcua, M.; Mata-Rocha, M.; Munoz-Torrico, M.; Salazar-Lezama, M. A.; Martinez-Orozco, J. A.; Narvaez-Diaz, L. A.; Salas-Hernandez, J.; Gonzalez-Covarrubias, V.; Soberon, X., A bioinformatics pipeline for Mycobacterium tuberculosis sequencing that cleans contaminant reads from sputum samples. PLoS One 2021, 16, (10), e0258774.
- de Welzen, L.; Eldholm, V.; Maharaj, K.; Manson, A. L.; Earl, A. M.; Pym, A. S., Whole-Transcriptome and -Genome Analysis of Extensively Drug-Resistant Mycobacterium tuberculosis Clinical Isolates Identifies Downregulation of ethA as a Mechanism of Ethionamide Resistance. Antimicrob Agents Chemother 2017, 61, (12).
- Gomez-Gonzalez, P. J.; Andreu, N.; Phelan, J. E.; de Sessions, P. F.; Glynn, J. R.; Crampin, A. C.; Campino, S.; Butcher, P. D.; Hibberd, M. L.; Clark, T. G., An integrated whole genome analysis of Mycobacterium tuberculosis reveals insights into relationship between its genome, transcriptome and methylome. Sci Rep 2019, 9, (1), 5204.
- Gupta, A.; Alland, D., Reversible gene silencing through frameshift indels and frameshift scars provide adaptive plasticity for Mycobacterium tuberculosis. Nat Commun 2021, 12, (1), 4702.
- Tang, J.; Liu, Z.; Shi, Y.; Zhan, L.; Qin, C., Whole Genome and Transcriptome Sequencing of Two Multi-Drug Resistant Mycobacterium tuberculosis Strains to Facilitate Illustrating Their Virulence in vivo. Front Cell Infect Microbiol 2020, 10, 219.
- My question regarding the preparation of Mtb cultures for infection and the control strains remains open. In their response, the authors presented experimental data confirming the same efficacy of infection for both strains (AFB/infected macrophages data). But my concern was this: because of the different growth conditions (Sauton medium for the control strains and 7H9 medium for the strains used for infection), the strains acquired different metabolic characteristics BEFORE infection. That could influence the list of DEGs dramatically.
Answer: Upon phagocytosis, Mtb initially resides into phagosomes. Different mechanisms have been shown to be used by Mtb to inhibit the consequences of the phagosome fusion to acidic vesicles, such as lysosomes. However, most studies have been focused on the host cells genes and little is known about the gene expression of mycobacteria, including Mtb and BCG, inside the phagosome, mostly due to the technical difficulties to isolate purified phagosomes and the limited amounts of Mtb residing in this organelle (He et al., 2012, J Proteome Res; Hoffmann et al., 2018 Front Immunol). Therefore, a few studies have addressed the transcriptomic and proteomic profiling of Mtb residing into the phagosome or simulating conditions observed in the Mtb phagosome (Fisher et al., 2002 J Bacteriol; Schnappinger et al., 2003 J Exp Med; Rohde et al., 2007 Cell Host Microbe; Singhal et al., 2016 Protein Pept Lett). In all cases, the genes expressed by the phagosome-residing mycobacteria were compared with the genes expressed by mycobacteria grown in axenic 7H9 or Sauton’s media.
As we previously answered to the interesting observation pointed by the Reviewer, macrophages were infected with Mtb coming from 7H9 cultures. Once infected, the bacteria will be sensing the restricted environment posed by the macrophage phagosome in the absence of replication. The few studies of Mtb gene expression inside phagosomes of macrophages indicate an adaption of Mtb to this environment which has been proposed as nutrient-restricted. In order to simulate such a nutrient-restricted environment, we took 7H9-cultured Mtb and cultured it for 6 hours in Sauton's minimal medium since this medium is nutrient-restricted compared with 7H9. Then, we compared the transcriptome expressed by intracellular Mtb with the transcriptome of Mtb cultured for 6 hours in Sauton's medium. To our knowledge, just a few studies have addressed the response of Mtb to the phagosome conditions imposed by macrophages, either transcriptomic or proteomic, but no metabolic reconstruction in these conditions have been so far published.
To give more clarity to the Reviewer, the experimental strategy used by us for this specific issue is represented as follows:
No answer for my point 4 was given.
Answer: Noticing that our initial response posed in the journal portal lacked this particular answer, we sent to the Editor an archive containing it to be sent to the Reviewer. Thus, we replicate the answer that was sent:
We are aware of this apparent contradiction. However, in the results presented in Baena and colleagues' (2019) paper, the differentially expressed genes (DEGs) were obtained compared to the genes expressed by M. tuberculosis H37Rv as a control in axenic culture conditions. In our submitted manuscript, as described in the corresponding section of Materials and Methods, DEGs were obtained after infection of macrophages in vivo, compared with Mtb grown for 6 hours in a minimal medium (Sauton). Therefore, the results were obtained under different experimental conditions (axenic vs. in vivo). Nevertheless, in both cases, both transcriptomic analysis indicates clear differences between both strains of Mb, either in response to axenic culture conditions or the in vivo infection.
References:
Baena, A.; Cabarcas, F.; Alvarez-Eraso, K. L. F.; Isaza, J. P.; Alzate, J. F.; Barrera, L. F., Differential determinants of virulence in two Mycobacterium tuberculosis Colombian clinical isolates of the LAM09 family. Virulence 2019, 10, (1), 695-710.
Fisher, M. A.; Plikaytis, B. B.; Shinnick, T. M., Microarray analysis of the Mycobacterium tuberculosis transcriptional response to the acidic conditions found in phagosomes. J Bacteriol 2002, 184, (14), 4025-32.
He, Y.; Li, W.; Liao, G.; Xie, J., Mycobacterium tuberculosis-specific phagosome proteome and underlying signaling pathways. J Proteome Res 2012, 11, (5), 2635-43.
Hoffmann, E.; Machelart, A.; Song, O. R.; Brodin, P., Proteomics of Mycobacterium Infection: Moving towards a Better Understanding of Pathogen-Driven Immunomodulation. Front Immunol 2018, 9, 86.
Rohde, K. H.; Abramovitch, R. B.; Russell, D. G., Mycobacterium tuberculosis invasion of macrophages: linking bacterial gene expression to environmental cues. Cell Host Microbe 2007, 2, (5), 352-64.
Schnappinger, D.; Ehrt, S.; Voskuil, M. I.; Liu, Y.; Mangan, J. A.; Monahan, I. M.; Dolganov, G.; Efron, B.; Butcher, P. D.; Nathan, C.; Schoolnik, G. K., Transcriptional Adaptation of Mycobacterium tuberculosis within Macrophages: Insights into the Phagosomal Environment. J Exp Med 2003, 198, (5), 693-704.
Singhal, N.; Kumar, M.; Sharma, D.; Bisht, D., Comparative Protein Profiling of Intraphagosomal Expressed Proteins of Mycobacterium bovis BCG. Protein Pept Lett 2016, 23, (1), 51-4.

Reviewer 3 Report
The paper improved but the quality of the figures is still an issue.
Author Response
Answer: The last archive we sent with the main figures was at 600 points/cm resolution. The interactomes provided by String (svg), were downloaded with the maximum resolution the program provides. We will try to get a higher resolution.
Round 3
Reviewer 2 Report
accept after the 2nd round